# Association between Atmospheric Particulate Pollutants and Mortality for Cardio-Cerebrovascular Diseases in Chinese Korean Population: A Case-Crossover Study

**DOI:** 10.3390/ijerph15122835

**Published:** 2018-12-12

**Authors:** Chao Zhang, Zhenyu Quan, Qincheng Wu, Zhezhen Jin, Joseph H. Lee, Chunhua Li, Yuxin Zheng, Lianhua Cui

**Affiliations:** 1School of Public Health, Medical College of Qingdao University, Qingdao 266021, Shandong Province, China; qduzec@163.com (C.Z.); wqc12345@163.com (Q.W.); yxzheng@qdu.edu.cn (Y.Z.); 2Medical School of Yanbian University, Yanji City 133002, Yanbian Korean Autonomous Prefecture, Jilin, China; zyquan@ybu.edu.cn; 3Department of Biostatistics, Mailman School of Public Health, Columbia University, New York, NY 10032, USA; zj7@cumc.columbia.edu; 4Sergievsky Center, Taub Institute, and Department of Epidemiology, Mailman School of Public Health, Department of Neurology, College of Physicians and Surgeons, Columbia University, New York, NY 10032, USA; JHL2@cumc.columbia.edu; 5Yanbian Korean Autonomous Prefecture Center for Disease Control and Prevention, Yanji City 133000, Yanbian Korean Autonomous Prefecture, Jilin, China; qduqcw@163.com

**Keywords:** PM_2.5_, PM_10_, cardiovascular disease, cerebrovascular disease

## Abstract

Background: Air pollution in large Chinese cities has led to recent studies that highlighted the relationship between particulate matters (PM) and elevated risk of cardio-cerebrovascular mortality. However, it is unclear as to whether: (1) The same adverse relations exist in cities with relatively low levels of air pollution; and (2) the relationship between the two are similar across ethnic groups. Methods: We collected data of PM_2.5_ (PM with an aerodynamic diameter ≤ 2.5 µm) and PM_10_ (aerodynamic diameter ≤ 10 µm) in the Yanbian Korean Autonomous Prefecture between 1 January 2015 and 31 December 2016. Using a time-stratified case-crossover design, we investigated whether levels of particulate pollutants influence the risk of cardio-cerebrovascular disease mortality among ethnic Korean vs. ethnic Han residents residing in the Yanbian Korean Autonomous Prefecture. Results: Under the single air pollutant model, the odds ratios (ORs) of cardio-cerebrovascular disease were 1.025 (1.024–1.026) for each 10 μg/m^3^ increase in PM_2.5_ at lag0 day, 1.012 (1.011–1.013) for each 10 μg/m^3^ increase in PM_10_ at lag1 day. In the multi-pollutant model adjusted by PM_10_, SO_2_, and NO_2_, the ORs of cardio-cerebrovascular disease were 1.150 (1.145–1.155) for ethnic Koreans and 1.154 (1.149–1.158) for ethnic Hans for each 10 μg/m^3^ increase in PM_2.5_. In the multi-pollutant model adjusted by PM_2.5_, SO_2_, and NO_2_, the ORs of cardio-cerebrovascular disease were 1.050 (1.047–1.053) for ethnic Koreans and 1.041 (1.039–1.043) for ethnic Hans for each 10 μg/m^3^ increase in PM_10_. Conclusion: This study showed that PM_2.5_ and PM_10_ were associated with increased risks of acute death events in residential cardio-cerebrovascular disease in Yanbian, China.

## 1. Introduction

Air pollution has been demonstrated to be a major risk factor for the development of cardiovascular diseases worldwide. A large number of studies have shown that air pollutants have an effect on cardiovascular morbidity and mortality [1,2,3,4,5]. Many epidemiological studies have showed that air pollution is a complex mixture of gaseous particulate compounds and that air pollution has negative health effects [6,7,8,9,10,11]. Experiments have indicated that particulate matter with a diameter ≤ 2.5 µm (PM_2.5_), a very small particle, can enter the body through the trachea, and then into the alveolar of lung tissue, and may spread to the capillaries and into the blood [12]. The PM_2.5_ can also contain carcinogens such as polycyclic aromatic hydrocarbons (PAHs) [13,14,15,16]. Thus, PM_2.5_ might have a serious negative impact on human health.

Studies in China on PM_2.5_ exposure have indicated that elevated levels of PM_2.5_ exposure can lead to an increased risk of cardiovascular mortality [17,18]. However, all of these studies have focused on the areas where PM_2.5_ exposure was higher than the national secondary standard (35 μg/m^3^).

On the other hand, numerous studies in Europe and America have focused on areas where the range of PM_2.5_ exposure was lower than 35 μg/m^3^ (specifically 6.7–25 μg/m^3^), and showed that the increase in PM_2.5_ levels can also lead to an increase in the risk of cardiovascular mortality [19,20,21,22]. Recently, a meta-analysis [16] summarized the impact of long-term exposure to PM_2.5_ on the mortality of cardio-respiratory disease in different populations. Eight of these studies showed that an increased PM_2.5_ exposure was associated with an increased risk of cardiovascular death among residents. In these studies, the lowest level of PM_2.5_ exposure was 9 μg/m^3^ and the highest was 23 μg/m^3^.

Studies conducted in other regions have shown that PM_10_ concentrations were associated with cardiovascular mortality as well [23,24,25]. A study conducted by Kan and colleagues [26] in Shanghai showed that for every 10 μg/m^3^ increase in PM_10_, the mortality rate of cardiovascular death in residents increased by 0.27%.

Yanbian Korean Autonomous Prefecture is a city bordering North Korea in Northeastern China with an area of 43,474 square kilometers and has a population of 2.177 million, of which the ethnic Korean Chinese accounts for 37.7%. The region’s PM_2.5_ annual average at 32.90 μg/m^3^. To our knowledge, this is the first study to investigate whether the PM_2.5_ exposure level in the city of Yanbian North Korea Autonomous Prefecture is related to the risk of cardiovascular mortality. Further, we will explore the difference in cardio-cerebrovascular risk associated with the PM_2.5_ exposure between ethnic Han Chinese (ethnic Hans) and ethnic Korean Chinese (ethnic Koreans) in the city of Yanbian Korean Autonomous Prefecture.

In this study, we used a time-stratified case-crossover design [20] to test the relationship between various particulate pollutants and cardio-cerebrovascular disease mortality. Since each case serves as its own control, confounding factors by invariant and slowly changing risk factors such as age, gender, race, smoking, genetic, and socio-economic status are controlled by study design itself.

## 2. Materials and Methods

### 2.1. Meteorological Data

All subjects gave their informed consent before they participated in the study. The study protocol was approved by Yanbian university medical college ethics committee (Project identification code 201839). We obtained routine monitoring data on air pollutants from the Environmental Monitoring Center of the Yanbian Korean Autonomous Prefecture. These included daily average concentrations of PM_2.5_, PM_10_, SO_2_, and NO_2_ from 1 January 2015 and 31 December 2016. In addition, we collected the daily mean temperature, relative humidity, and barometric pressure from the Yanbian Korean Autonomous Prefecture Meteorological Bureau.

### 2.2. Mortality Data

For the same time period, we obtained mortality data related to cardiovascular and cerebrovascular diseases along with demographic data (e.g., age, sex, date of death, and address) from the Yanbian Korean Autonomous Prefecture Center for Disease Control and Prevention. The classification of the disease in the death cases was coded according to the International Classification of Diseases (ICD-10). For the present study, ICD-10 codes associated with cardiovascular disease (I09–I11, I20–I27, and I47–I52) and cerebrovascular disease (I60–I69) were included. The present study is restricted to ethnic Koreans and ethnic Hans, because there were limited data on other ethnic groups.

### 2.3. Data Analysis

The case-crossover design was developed as a variant of the case-control design to study effects of transient exposures on acute events [27,28]. The method can be regarded as a special type of case-control study in which each case serves as his/her own control. Since there is nearly perfect matching on all measured or unmeasured personal characteristics that do not vary over time, there can be no confounding by those characteristics. The method uses cases with cardiovascular or cerebrovascular diseases as their own controls, thereby effectively controlling all factors (including age, gender, race, smoking, genetic, and socio-economic status) except for the exposure of interest. We thus applied the time-stratified case-crossover design to assess relations between levels of air pollutants and the risk of cardiovascular and cerebrovascular deaths. The case period for each death was defined as the day of death. Control periods were selected by matching the same day of the week within the same calendar month and year for each case. For example, if a case was on the second Thursday of April 2016 (14 April 2016), all other Thursdays within April 2016 were assigned as controls (7 April 21, and 28 April 2016). This approach resulted in three or four controls for each case. This approach allows unbiased conditional logistic regression estimates [29,30]. Previous studies have shown that there is a lagging effect of air pollution on death [31]. Therefore, this study used to model lagging to observe the impact of PM_2.5_ and PM_10_ on the day (lag0) and delay 1–3 days (lag1–3) of death for cardio-cerebrovascular disease in residents, respectively. Lag1 is defined as the pollutant exposure value of the one day before the deaths, lag2 as the pollutant exposure value for two days before deaths, and so on.

The relations among the measures of PM_2.5_, PM_10_, SO_2_, and NO_2_ were assessed by Pearson correlation analysis. The effects of atmospheric PM_2.5_ and PM_10_ on the exposed population were summarized by the difference between the health effects of each layer and its 95% confidence interval based on the formula: $$(Q_{1}-Q_{2})\pm 1.96\;\sqrt{{\left(\mathrm{SE}1\right)}^{2}+{\left(\mathrm{SE}2\right)}^{2}}$$
where Q_1_ and Q_2_ were the estimates of the two health effects and SE1 and SE2 were their corresponding standard errors.

The conditional logistic regression analysis was used to examine the risk of the single-pollutant and multi-pollutant exposure conditions, and atmospheric PM_2.5_ and PM_10_ exposures in the case period and control period on cardio-cerebrovascular disease deaths. To take into account the impact of meteorological factors on cardio-cerebrovascular mortality, the average daily temperature, relative humidity and barometric pressure were included in the regression model. Subgroup analysis was also carried out with the categorized factors of sex, age, and season: Sex (male vs. female), age (≤ 65 years vs. > 65 years), and season (cold vs. warm season). Warm season was defined as May to October, cold season was defined as November to April. Odds ratios (ORs) for PM_2.5_ and PM_10_ were computed for each unit increase of 10 μg/m^3^ and 95% confidence intervals (CI) were provided. Data analysis was performed with SPSS 17.0 software (SPSS Inc, Chicago, IL, USA) and all statistical tests were two-sided at the significance level 0.05.

## 3. Results

The total number of deaths from cardio-cerebrovascular disease among the residents of Yanbian Korean Autonomous Prefecture during the study period was 16,365 cases (9029 ethnic Hans, and 7336 ethnic Koreans): 8040 deaths due to cardiovascular disease; 8325 deaths due to cerebrovascular disease; 8998 male deaths and 7367 female deaths. Among the 8998 male deaths, 4160 were due to cardiovascular disease. Among the 7367 female deaths, 3880 were due to cardiovascular disease (Table 1). On average, there were approximately 22.4 cardio-cerebrovascular disease deaths per day. The average concentrations of PM_2.5_, PM_10_, NO_2_, SO_2_, temperature, relative humidity, and barometric pressure were 32.90 μg/m^3^, 51.20 μg/m^3^, 22.30 μg/m^3^, 14.75 μg/m^3^, 5.46 °C, 66.09%, and 962.46 hPa, respectively. The average levels of PM_2.5_, PM_10_, SO_2_, and NO_2_ were lower than the national secondary ambient air quality standard in China. The national secondary ambient air quality standard in China of PM_2.5_, PM_10_, SO_2_, and NO_2_ were 35 μg/m^3^, 70 μg/m^3^, 60 μg/m^3^, and 40 μg/m^3^, respectively. The mean value of each pollutant in the cold season was higher than the mean value of each pollutant in the warm season (Table 2). Pearson correlation analysis on PM_2.5_, PM_10_, SO_2_, NO_2_, temperature, relative humidity, and barometric pressure showed that air pollutants were significantly positively correlated with barometric pressure, but were significantly negatively correlated with temperature and relative humidity. As shown in Table 3, increased levels of PM_2.5_ and PM_10_ yielded significant increases in the cardio-cerebrovascular disease mortality. The ORs were 1.025 (1.024–1.026) for each 10 μg/m^3^ increase in PM_2.5_ at lag0 day, 1.012 (1.011–1.013) for each 10 μg/m^3^ increase in PM_10_ at lag one day. Based on single pollutant model, multi-pollutant models were built for the ethnic Koreans and ethnic Hans, respectively (Table 4). In the multi-pollutant model adjusted by PM_10_, SO_2_, and NO_2_, the ORs of cardio-cerebrovascular disease were 1.150 (1.145–1.155) for ethnic Koreans and 1.154 (1.149–1.158) for ethnic Hans for each 10 μg/m^3^ increase in PM_2.5_. In the multi-pollutant model adjusted by PM_2.5_, SO_2_, and NO_2_, the ORs of cardio-cerebrovascular disease were 1.050 (1.047–1.053) for ethnic Koreans and 1.041 (1.039–1.043) for ethnic Hans for each 10 μg/m^3^ increase in PM_10_. The results indicate that the effect of PM_2.5_ on cerebrovascular disease mortality were slightly higher for ethnic Hans than that for ethnic Koreans. While the multi-pollutant models showed that the effect of adjusted PM_10_ on the risk of cardiovascular and cerebrovascular deaths in ethnic Koreans were greater than in ethnic Hans (Table 4). Table 5 showed the results from subgroup analyses by gender, age, and season. It showed that there was difference in results with the seasonal stratification, but no difference in results with the gender and age stratifications. Table 6 summarizes available results from studies similar to our study, which shows a consistent significant risk of PM_2.5_ and PM_10_ on cardiovascular mortality.

## 4. Discussion

Our study showed that levels of PM_2.5_ were significantly associated with cardiovascular mortality in the Yanbian Korean Autonomous Prefecture, even though the daily mean value of PM_2.5_ concentration was lower than China’s national secondary standard of 35 μg/m^3^ for PM_2.5_ during the study period. To our knowledge, this is the first study on the relationship between PM_2.5_ and the death of cardiovascular disease in China with the data that the PM_2.5_ exposure concentration is lower than that of the national secondary standard. In addition, this study extends the significant public health findings by investigating whether the relationship between PM_2.5_ and cardiovascular mortality differs by ethnicity. In our study, a 10 μg/m^3^ increment in the lag one day concentration of PM_2.5_ corresponded to 2.6% (95%CI: 2.5%, 2.8%) increase of cardiovascular mortality for total population in Yanbian Korean Autonomous Prefecture. Our estimates of cardiovascular mortality in the Yanbian Korean Autonomous Prefecture were higher than those in studies of Guo [39], Ma [18], and Venners [40] in other parts of China, including Beijing, Shenyang, and Chongqing, respectively. Our results were similar to those from studies conducted in developed countries with lower levels of PM_2.5_ that showed a significant impact on cardiovascular mortality [22,32,33]. A possible explanation of this phenomenon could lie in the exposure–response curve of air pollution that often tends to become flat at higher concentration levels [41].

In the single pollutant model, the results show that PM_2.5_ is more harmful than PM_10_ for the entire population residing in the Yanbian Korean Autonomous Prefecture, which was similar to the results of Guo’s [42] and Dai’s [43]. This might be due to the fact that PM_2.5_ has a smaller particle size and has a larger specific surface area, so it can absorb more harmful substances than PM_10_, and PM_2.5_ can stay longer in the air. In addition, PM_2.5_ can be inhaled directly into the alveoli, participate in blood circulation, and may ultimately have a direct impact on the cardiovascular system.

It is interesting to see that ORs of PM_2.5_ and PM_10_ in the multi-pollutant models were higher than that in the single pollutant model in ethnic Korean Chinese and ethnic Han Chinese. Additionally, the effects of adjusted PM_10_ on the risk of cardiovascular and cerebrovascular deaths in ethnic Koreans were greater than those of ethnic Hans (Table 4). This may be due to the combined effects of particulate contaminants and gaseous pollutants on cardio-cerebrovascular disease. Our analysis also showed that there might be an interaction between PM_2.5_ and PM_10_, and both have synergistic effects on the health of the human body.

In addition, the effect of PM_2.5_ in ethnic Hans had slightly higher risk for cerebrovascular disease mortality compared to the ethnic Koreans (Table 4). Similarly, Ostro and colleagues [21], using data from nine California Counties, reported that elevated PM_2.5_ in Caucasians was associated with a higher risk of cardio-respiratory disease mortality compared with the cardio-respiratory disease mortality in African Americans. Potentially, this may be due to the differences in dietary habits and genetic factors between ethnic groups, but further studies are needed.

Our subgroup analysis showed that the effects of PM_2.5_ and PM_10_ on the cardio-cerebrovascular mortality significant differed by seasons. The PM_2.5_ and PM_10_ in the warm season have higher impact on cardio-cerebrovascular diseases mortality than those in the cold season. It is consistent with the results from the work of Ma and colleagues in Shenyang [18]. It might be due to the fact that temperature is higher in the warm season and PM_2.5_ is more likely to diffuse in the air, resulting in more inhalation of the body and eventually into the human blood circulation. There is no consensus on the effects of seasonal factors on the relationship between atmospheric PM_10_ exposure and population death, as demonstrated by Peng [44], Nawrot [45], and Lu [46]. These studies have shown that the warm season PM_10_ pollution has a higher impact on population mortality than the cold season. It is believed that air pollution affects the biological pathway of cardiovascular mechanism [47,48], but it is not clear in which way.

Our study observed no significant differences in health effects of PM_2.5_ and PM_10_ exposure on the cardio-cerebrovascular mortality when stratified by age and gender. It is consistent with the results of the studies performed in Guangzhou and Shanghai by Kan and colleagues [34,49]. However, Franklin and colleagues [32], using data from 27 US communities, reported that an increase in PM_2.5_ concentration had a stronger impact for cardiovascular mortality in women and seniors (75 years or older). Hong and colleagues [50] also found that elderly women were most susceptible to the adverse effects of PM_10_ on the risk of acute mortality from stroke. The difference might be due to the difference in study population and in the criteria for inclusion, the number of cases, age stratification, composition of air pollutants, and degree of contamination. Further studies are needed.

Finally, comparison to the studies similar to ours showed higher PM_2.5_ cardiovascular mortality in the Yanbian Korean Autonomous Prefecture compared to those in Shenyang, Shijiazhuang, Wuhan, Guangzhou, and Shanghai in China. In previous studies, it appeared that PM_2.5_ exposure levels in developed countries were lower than in many Chinese cities, and some cities were even lower than China’s national secondary standards. However, the PM_2.5_ cardiovascular mortalities in those countries were higher than those in Chinese studies. It is interesting to see that the increase in cardiovascular disease death caused by PM_2.5_ exposure in the Yanbian Korean Autonomous Prefecture was also higher than that in other cities in China. One possible explanation might lie in different research backgrounds, such as local PM levels, population sensitivity to PM_2.5_, age structure, and particle composition and toxicity [17]. Another possibility might be based on a reasoning that, at high exposure concentrations, the risk of death per unit increase of pollutant concentrations often tends to be reduced, possibly because vulnerable subjects may have died before the concentration had reached the maximum level [51]. In addition, the PM_10_ cardiovascular mortality in our study was higher than those in studies in Shanghai and Sao Paulo, but was lower than that in a European study [26,35,36]. These inconsistent findings might be due to the regional differences in the study, such as the population living at different latitudes and the sensitivity of the population to PM, which is of interest for further investigation.

The research has some limitations. The data we collected was limited to only one city with a limited sample size and study duration period. While our study has showed that the increase in the concentration of atmospheric particulate pollutants would increase the risk of cardio-cerebrovascular mortality in the Yanbian Korean Autonomous Prefecture, the causal relationship between the two is unclear. Further investigation is needed to explore the biological mechanism.

## 5. Conclusions

In summary, this study shows that airborne particulate pollutants can increase the risk of acute death events in residential cardiovascular disease and cerebrovascular disease in the Yanbian Korean Autonomous Prefecture. The synergies between the various pollutants will increase their impact on cardio-cerebrovascular mortality, especially the combination of PM_2.5_ and PM_10_ will lead to a significant increase in cardio-cerebrovascular mortality. The cerebrovascular disease mortality risk associated with PM_2.5_ was slightly higher in ethnic Hans than in ethnic Koreans. However, due to the small magnitude of the difference, further studies are needed to confirm and elaborate the findings.

## Figures and Tables

**Table 1 ijerph-15-02835-t001:** Distribution of cardio-cerebrovascular diseases for total population in Yanbian, China.

Ethnic		Cardio-Cerebrovascular Disease Number (Percentage) ^a^	Cardiovascular Diseases Number (Percentage)	Cerebrovascular Disease Number (Percentage)
Han		9029 (55.2%)	4785 (29.2%)	4244 (26.0%)
	Male	5289 (32.3%)	2662 (16.3%)	2627 (16.1%)
	Female	3740 (22.9%)	2123 (12.9%)	1617 (9.9%)
Korean		7336 (44.8%)	3255 (19.9%)	4081 (24.9%)
	Male	3709 (22.7%)	1498 (9.2%)	2211 (13.5%)
	Female	3627 (22.1%)	1757 (10.7%)	1870 (11.4%)
Total		16,365 (100%)	8040 (49.1%)	8325 (50.9%)

^a^ The number outside the brackets is the number of deaths from the disease, with the figures in brackets as the percentage of deaths.

**Table 2 ijerph-15-02835-t002:** Levels of daily air pollutants and deaths from cardio-cerebrovascular disease in Yanbian, China.

	Minimum	Maximum	Mean ± SD ^a^	Cold Season ^b^ (mean)	Warm Season ^c^ (mean)
PM_2.5_ (μg/m^3^)	4	289	32.90 ± 28.27	48.41	17.80
PM_10_ (μg/m^3^)	8	285	51.20 ± 35.82	68.46	34.42
NO_2_ (μg/m^3^)	6	88	22.30 ± 12.04	27.27	17.47
SO_2_ (μg/m^3^)	1	128	14.75 ± 16.63	25.55	4.24
Temperature (°C)	−24	28	5.46 ± 12.87	−5.16	15.79
Relative humidity (%)	15	97	66.09 ± 15.70	59.86	72.15
Barometric (hPa)	932	982	962.46 ± 7.35	966.24	958.80
Total deaths	9	40	22.39 ± 5.14	23.70	21.09

^a^ SD standard deviation; ^b^ Warm season defined as May to October; ^c^ Cold season defined as November to April. PM_2.5:_ particulate matter with a diameter ≤ 2.5 µm; PM_10:_ particulate matter with a diameter ≤ 10 µm.

**Table 3 ijerph-15-02835-t003:** OR value of cardio-cerebrovascular disease mortality associated with 10 μg/m^3^ increase of PM_2.5_ and PM_10_ concentrations.

Air Pollutants	Lag Time (days) ^a^	Cardio-Cerebrovascular Diseases OR (95%CI) ^b^	Cardiovascular Diseases OR (95%CI)	Cerebrovascular Disease OR (95%CI)
PM_2.5_	0	1.025 * (1.024–1.026) ^c^	1.025 * (1.023–1.026)	1.024 * (1.023–1.026) ^c^
	1	1.025 * (1.024–1.026)	1.026 * (1.025–1.028) ^c^	1.024 * (1.023–1.025)
	2	1.024 * (1.023–1.025)	1.023 * (1.022–1.025)	1.024 * (1.023–1.026)
	3	1.023 * (1.022–1.024)	1.024 * (1.022–1.025)	1.023 * (1.022–1.024)
PM_10_	0	1.012 * (1.011–1.012)	1.012 * (1.011–1.013) ^c^	1.011 * (1.010–1.012)
	1	1.012 * (1.011–1.013) ^c^	1.012 * (1.011–1.013)	1.012 * (1.011–1.013) ^c^
	2	1.011 * (1.011–1.012)	1.011 * (1.010–1.011)	1.012 * (1.011–1.013)
	3	1.010 * (1.010–1.011)	1.010 * (1.009–1.011)	1.010 * (1.009–1.011)

* *p* < 0.05; ^a^ Lag 0 the pollutant exposure value for the day of the deaths, Lag 1 the pollutant exposure value of the one day before the deaths, Lag 2 the pollutant exposure value of the two days before the deaths, and Lag 3 the pollutant exposure value of the three days before the death; ^b^ OR odds ratio, CI confidence interval; ^c^ The highest health effects OR value of the pollutants in lag 0–3 days. PM_2.5_: particulate matter with a diameter ≤ 2.5 µm; PM_10_: particulate matter with a diameter ≤ 10 µm.

**Table 4 ijerph-15-02835-t004:** OR value of cardio-cerebrovascular disease mortality associated with 10 μg/m^3^ increase of PM_2.5_ and PM_10_ in multiple-pollutant models for ethnic Korean Chinese and ethnic Han Chinese in Yanbian North Korea Autonomous Prefecture, China.

Air Pollutants	Cardio-Cerebrovascular Disease		Cardiovascular Disease		Cerebrovascular Disease	
	Han ^a^ OR (95%CI) ^c^	Korean ^b^ OR (95%CI)	Han OR (95%CI)	Korean OR (95%CI)	Han OR (95%CI)	Korean OR (95%CI)
PM_2.5_	1.025 * (1.024–1.027)	1.024 * (1.022–1.025)	1.027 * (1.025–1.029)	1.026 * (1.023–1.028)	1.026 * (1.024–1.028) ^d^	1.023 * (1.021–1.025) ^d^
+PM_10_	1.142 * (1.137–1.146)	1.137 * (1.132–1.142)	1.142 * (1.136–1.149)	1.139 * (1.132–1.147)	1.147 * (1.140–1.154)	1.147 * (1.140–1.154)
+SO_2_	1.040 * (1.038–1.042)	1.039 * (1.037–1.041)	1.041 * (1.038–1.044)	1.042 * (1.039–1.046)	1.040 * (1.037–1.043)	1.039 * (1.036–1.042)
+NO_2_	1.058 * (1.056–1.060)	1.058 * (1.055–1.060)	1.060 * (1.057–1.063)	1.061 * (1.057–1.065)	1.059 * (1.056–1.063)	1.057 * (1.053–1.060)
+SO_2_+NO_2_	1.059 * (1.057–1.061)	1.059 * (1.056–1.061)	1.060 * (1.057–1.064)	1.062 * (1.058–1.067)	1.060 * (1.057–1.064)	1.058 * (1.054–1.061)
+PM_10_+NO_2_+SO_2_	1.154 * (1.149–1.158)	1.150 * (1.145–1.155)	1.157 * (1.150–1.164)	1.154 * (1.145–1.162)	1.158 * (1.151–1.165)	1.159 * (1.152–1.166)
PM_10_	1.012 * (1.011–1.013)	1.012 * (1.011–1.013)	1.012 * (1.011–1.013)	1.012 * (1.011–1.014)	1.012 * (1.010–1.013)	1.012 * (1.010–1.013)
+PM_2.5_	1.041 * (1.039–1.043) ^d^	1.050 * (1.047–1.053) ^d^	1.043 * (1.040–1.046) ^d^	1.049 * (1.044–1.053) ^d^	1.040 * (1.037–1.043) ^d^	1.048 * (1.044–1.052) ^d^
+SO_2_	1.015 * (1.014–1.016)	1.017 * (1.016–1.018)	1.016 * (1.015–1.018)	1.017 * (1.015–1.019)	1.015 * (1.013–1.016)	1.017 * (1.015–1.018)
+NO_2_	1.018 * (1.017–1.019)	1.020 * (1.019–1.022)	1.018 * (1.017–1.020) ^d^	1.021 * (1.019–1.023) ^d^	1.017 * (1.016–1.019) ^d^	1.020 * (1.018–1.022) ^d^
+SO_2_+NO_2_	1.018 * (1.017–1.019) ^d^	1.021 * (1.019–1.022) ^d^	1.019 * (1.017–1.021) ^d^	1.022 * (1.020–1.024) ^d^	1.018 * (1.016–1.019)	1.020 * (1.019–1.022)
+PM_2.5_+NO_2_+SO_2_	1.041 * (1.039–1.043) ^d^	1.050 * (1.047–1.053) ^d^	1.044 * (1.041–1.047) ^d^	1.050 * (1.045–1.054) ^d^	1.040 * (1.037–1.044) ^d^	1.049 * (1.045–1.053) ^d^

* *p* < 0.05; ^a^ Ethnic Han Chinese; ^b^ Ethnic Korean Chinese; ^c^ OR, odds ratio; CI, confidence interval; ^d^ The difference of effect estimate in the ethnic Korean Chinese and ethnic Korean Chinese was statistically significant (*p* < 0.05). PM_2.5_: particulate matter with a diameter ≤ 2.5 µm; PM_10_: particulate matter with a diameter ≤ 10 µm.

**Table 5 ijerph-15-02835-t005:** The association of cardio-cerebrovascular disease mortality with 10 µg/m^3^ increase of PM_2.5_/PM_10_ concentrations in different stratifications.

Air Pollutants		Cardio-Cerebrovascular Disease OR (95%CI) ^a^	Cardiovascular Diseases OR (95%CI)	Cerebrovascular Disease OR (95%CI)
Male	PM_2.5_	1.025 * (1.024–1.027)	1.025 * (1.023–1.027)	1.025 * (1.023–1.027)
	PM_10_	1.012 * (1.011–1.013)	1.012 * (1.011–1.013)	1.012 * (1.011–1.013)
Female	PM_2.5_	1.024 * (1.023–1.025)	1.027 * (1.025–1.030)	1.023 * (1.021–1.026)
	PM_10_	1.011 * (1.011–1.012)	1.012 * (1.011–1.013)	1.011 * (1.010–1.013)
Age ≤ 65 years	PM_2.5_	1.024 * (1.022–1.026)	1.026 * (1.023–1.029)	1.024 * (1.021–1.026)
	PM_10_	1.012 * (1.011–1.013)	1.011 * (1.009–1.013)	1.012 * (1.010–1.014)
Age > 65 years	PM_2.5_	1.025 * (1.024–1.026)	1.026 * (1.025–1.028)	1.025 * (1.023–1.026)
	PM_10_	1.012 * (1.011–1.013)	1.012 * (1.011–1.013)	1.012 * (1.011–1.013)
Warm season ^b^	PM_2.5_	1.141 * (1.136–1.145) ^d^	1.136 * (1.130–1.142) ^d^	1.139 * (1.133–1.146) ^d^
	PM_10_	1.038 * (1.036–1.040) ^e^	1.038 * (1.035–1.041) ^e^	1.036 * (1.033–1.039) ^e^
Cold season ^c^	PM_2.5_	1.014 * (1.013–1.015) ^d^	1.015 * (1.013–1.016) ^d^	1.014 * (1.012–1.015) ^d^
	PM_10_	1.008 * (1.007–1.009) ^e^	1.008 * (1.007–1.009) ^e^	1.008 * (1.007–1.009) ^e^

* *p* < 0.05; ^a^ OR odds ratio, CI confidence interval; ^b^ Warm season defined as May to October; ^c^ Cold season defined as November to April; ^d^ The difference of effect estimate in the cold and warm seasons was statistically significant for PM_2.5_ (*p* < 0.05); ^e^ The difference of effect estimate in the cold and warm seasons was statistically significant for PM_10_ (*p* < 0.05).

**Table 6 ijerph-15-02835-t006:** Estimates of percent increase in mortality for 10 μg/m^3^ increment of PM_2.5_/PM_10_ in multicity studies.

Study Locations	Reference	Study Period	Exposure	Cardiovascular Mortality
Our study	—	2015–2016	PM_2.5_ (32.9 ± 28.3)	2.60 (2.50, 2.80)
	—		PM_10_ (51.2 ± 35.8)	1.20 (1.10, 1.30)
Shenyang, China	[18]	2006–2008	PM_2.5_ (75 ± 43)	0.53 (0.09, 0.97)
Shanghai, China	[17]	2004–2005	PM_2.5_ (56.4 ± 1.3)	0.41 (0.01, 0.82)
U.S. Californian 9 cities	[21]	1999–2002	PM_2.5_ (19.4)	0.60 (0.02, 1.00)
U.S. 27 communities	[32]	1997–2002	PM_2.5_ (15.7)	1.03 (0.02, 2.04) ^a^
U.S. 112 cities	[22]	1999–2005	PM_2.5_ (6.7–25)	0.85 (0.46, 1.24)
Japan 20 cities	[33]	2002–2004	PM_2.5_ (11.8–22.8)	—
Guangzhou, China	[34]	2007–2008	PM_2.5_ (70.1 ± 34.6)	1.22 (0.63, 1.68)
Europe 30 cities	[35]	1990–1997	PM_10_ (22.5–76.2)	1.97 (1.38, 2.55)
Shanghai, China	[26]	2001–2004	PM_10_ (102 ± 65)	0.27 (0.10, 0.44)
Sao Paulo, Brazil	[36]	2000–2011	PM_10_ (40.8)	0.40 (0.07, 0.73)
Taiwan, China	[37]	2006–2008	PM_2.5–10_ (21.45)	10.0 (1.0, 21.0)
Shijiazhuang, China	[38]	2013–2015	PM_2.5_ (117 ± 99)	0.29 (0.10, 0.47)
Shanghai, China	[38]	2013–2015	PM_2.5_ (56 ± 38)	0.29 (0.04, 0.55)
Wuhan, China	[38]	2013–2015	PM_2.5_ (79 ± 55)	0.44 (0.05, 0.83)
Guangzhou, China	[38]	2013–2015	PM_2.5_ (46 ± 25)	1.42 (0.85, 2.00)

^a^ Stroke.

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
