# Peer review of "Association between Atmospheric Particulate Pollutants and Mortality for Cardio-Cerebrovascular Diseases in Chinese Korean Population: A Case-Crossover Study"

_ijerph, 2018, doi:10.3390/ijerph15122835_

Round 1
Reviewer 1 Report
The manuscript deals with the association between exposure to particulate matter and mortality for cardiovascular and cerebrovascular diseases. The topic is of great interest, especially given the generally high levels of atmospheric pollution in China regions, and would provide a real contribute to this field of research. Nonetheless, the paper needs to be improved.
In introduction, statements at lines 46-48 and at lines 48-50 are quite unclear. The authors need to rewrite both.
At line 58, do “literatures” stay for “studies”? Please explain that.
In addition, introduction should also contain references to studies related to the association between PM10 and increase in cardiovascular mortality.
In methods, the authors need to rewrite statements at lines 99-101, which are unclear: the “control” period is the timeframe during which the subject is considered as a control.
In results, please check the values relative to temperature, relative humidity and barometric pressure (lines 133-134). They seem incorrect.
The statement at lines 151-153 is incorrect in English. Please rewrite it.
Description of table 5 is incomplete. The authors should give a brief explanation on results regarding differences by age and sex.
In discussion, it is useful to clarify with appropriate examples whether there are specific differences (e.g., age, race, etc.) between the considered population in this study and those in previous investigations.
Finally, I recommend a revision of the text from an English mother-tongue. Please note that in introduction there is an excessive use of the verb “to show”.
Author Response
Dear Reviewer,
We are very grateful of your very constructive comments. According to your suggestions, we revised our manuscript. Below you will find our point-by-point responses to your comments.
Point 1: In introduction, statements at lines 46-48 and at lines 48-50 are quite unclear. The authors need to rewrite both.
Response 1: Thank you for your very important suggestion. We have revised the sentences as following:
“Experiments have indicated that PM2.5, a very small particle, can enter the body through the trachea, and then into the alveolar of lung tissue, and may spread to the capillaries and into the blood [12]. The PM2.5 also containing carcinogens such as polycyclic aromatic hydrocarbons (PAHs) [13-16]. Therefore, PM2.5 has a serious negative impact on human health.”
Point 2: At line 58, do “literatures” stay for “studies”? Please explain that.
Response 2: Thank you for your detailed review. We have changed “literatures” to “studies” in the manuscript (lines 66-67).
Point 3: In addition, introduction should also contain references to studies related to the association between PM10 and increase in cardiovascular mortality.
Response 3: Thank you for your constructive suggestion. We have added following sentences:
“Moreover, studies conducted in other regions have shown that PM10 concentrations are associated with cardiovascular mortality [23-25]. A study conducted by Kan and colleagues [26] in Shanghai showed that for every 10ug/m3 increase in PM10, the mortality rate of cardiovascular death in residents increased by 0.27%.”
Point 4: In methods, the authors need to rewrite statements at lines 99-101, which are unclear: the “control” period is the timeframe during which the subject is considered as a control.
Response 4: Thanks for your very important comments. Following your suggestion, we have revised the sentences as following:
“Control periods were selected by matching the same day of the week within the same calendar month and year for each case. For example, if a case was on the second Thursday of April 2016 (April 14, 2016), all other Thursdays within April 2016 were assigned as controls (April 7, 21, and 28, 2016). This approach resulted in three or four controls for each case.”
Point 5: In results, please check the values relative to temperature, relative humidity and barometric pressure (lines 133-134). They seem incorrect.
Response 5: Thank you for your detailed review again. According to your comment, we have corrected accordingly in the manuscript (lines 150-151).
Point 6: The statement at lines 151-153 is incorrect in English. Please rewrite it.
Response 6: Thank you for your constructive comment. According to your suggestion, we have revised the sentence as following:
“Based on single pollutant models, multi-pollutant models were built for the ethnic Korean Chinese and ethnic Han Chinese, respectively [see Appendix A]. In the multi-pollutant model adjusted by PM10, SO2 and NO2, the ORs of cardio-cerebrovascular disease were 1.150 (1.145-1.155) for Korean ethnic group and 1.154 (1.149-1.158) for Han ethnic group per 10ug/m3 increase in PM2.5. In the multi-pollutant model adjusted by PM2.5, SO2 and NO2, the ORs of cardio-cerebrovascular disease were 1.050(1.047-1.053) for Korean ethnic group and 1.041(1.039-1.043) for Han ethnic group per 10ug/m3 increase in PM10.” in the Abstract and Results.”
Point 7: Description of table 4 is incomplete. The authors should give a brief explanation on results regarding differences by age and sex.
Response 7: Thank you for your valuable advice. We have revised the sentence as following:
“Table 4 shows the results from subgroup analyses by gender, age and season. It shows that there was difference in results with the seasonal stratification, but no difference in results with the gender and age stratifications.”
Point 8: In discussion, it is useful to clarify with appropriate examples whether there are specific differences (e.g., age, race, etc.) between the considered population in this study and those in previous investigations.
Response 8: Thank you for your very constructive suggestions. Following your suggestions, we have added following sentences in the discussion:
“Similarly,Ostro et al. [21] using data from 9 California Counties reported that the effect of PM2.5 in white people had higher risk for cardio-respiratory disease mortality compared to the black people. It might be due to the differences in dietary habits and genetic factors among different ethnic groups,the specific factors need to be further studied.”
“In our study, no significant differences were observed in the effect of PM2.5 and PM10 exposure on cardio-cerebrovascular mortality between different age and gender groups. It is consistent with the results of Kan's [34, 49] studies in Guangzhou and Shanghai. However, Franklin and colleagues [32] using data from 27 US communities reported that increase in PM2.5 concentration had stronger impact for cardiovascular mortality in women and seniors (75 years or older). Hong et al. [50] also found that elderly women were most susceptible to the adverse effects of PM10 on the risk of acute mortality from stroke. The difference might be due to the difference in study population and in the criteria for inclusion, the number of cases, age stratification, composition of air pollutants, and degree of contamination. Further study is needed.”
Point 9: Finally, I recommend a revision of the text from an English mother-tongue. Please note that in introduction there is an excessive use of the verb “to show”.
Response 9: Thank you for your constructive comment again. Based on your suggestion, we have corrected accordingly in the manuscript.
Reviewer 2 Report
1. P3, L115: I’m not sure that the conditional logistic regression analysis is same as the Cox regression analysis.
2. The levels of air pollutants of the control periods, which could be included in Table 2.
3. Some limitations should be mentioned.
Author Response
Dear Reviewer,
We are very grateful of your very constructive comments. According to your suggestions, we revised our manuscript. Below you will find our point-by-point responses to your comments.
Point 1: P3, L115: I’m not sure that the conditional logistic regression analysis is same as the Cox regression analysis.
Response 1: We are very grateful of your detailed review, and sorry for the confusion. In SPSS and SAS softwares, the conditional logistic regression models are fit through the fit of the proportional Cox regression models. To avoid confusion, we have deleted the “i.e., the Cox regression analysis” in the manuscript (line 131).
Point 2: The levels of air pollutants of the control periods, which could be included in Table 2.
Response 2: Thank you for your valuable advice. Based on your suggestion, we have made the following explanations:
“The time-stratified case crossover study used in this study is different from the comparative study of two independent samples. Using this design, each subject serves as his or her own control by using the exposure period before or after the index date. The case period for each death was defined as the day of death. Control periods were selected by matching the same day of the week within the same calendar month and year for each case. For example, if a case was on the second Thursday of April 2016 (April 14, 2016), all other Thursdays within April 2016 were assigned as controls (April 7, 21, and 28, 2016). Similarly, if a case was on the third Thursday of April 2016 (April 21, 2016), all other Thursdays within April 2016 were assigned as controls (April 7, 14, and 28, 2016). Therefore, there was no fixed control period in this study, the concentration of pollutants in the control period could not be calculated.”
Point 3: Some limitations should be mentioned.
Response 3: Thank you for your constructive suggestion. We have added following sentences in the discussion: “The research has some limitations. The data we collected was limited to only one city with a limited sample size and study duration period. Although our study has showed that the increase in the concentration of atmospheric particulate pollutants would increase the risk of cardio-cerebrovascular mortality in the Yanbian Korean Autonomous Prefecture, the causal relationship between the two is unclear. Further investigation is needed to explore the biological mechanism.”
Round 2
Reviewer 1 Report
The manuscript has been significantly improved. However, presentation of results can be further ameliorated.
I do not understand why the important results relative to the multi-pollutant models were reported in Appendix A. In my opinion, these table needs to be included in results section.
In addition, there is only a slight increased risk of cardio-cerebrovascular disease for Han ethnic group per 10µg/m3 increase in PM2.5 in comparison to Korean ethnic group. I suggest the authors to rewrite the statements about this association. Are the authors confident that considering the probability of error, are the risk estimates not the same? Given this, the conclusions should also be reformulated (lines 267-268).
Author Response
Dear Reviewer,
We are very grateful of your very constructive comments. According to your suggestions, we revised our manuscript. Below you will find our point-by-point responses to your comments.
Point 1: I do not understand why the important results relative to the multi-pollutant models were reported in Appendix A. In my opinion, these table needs to be included in results section.
Response 1: Thank you for your constructive comment. Following your suggestion, we have included the multi-pollutant models into the results section, and changed “Appendix A” to “Table 4” in the manuscript.
Point 2: In addition, there is only a slight increased risk of cardio-cerebrovascular disease for Han ethnic group per 10µg/m3 increase in PM2.5 in comparison to Korean ethnic group. I suggest the authors to rewrite the statements about this association. Are the authors confident that considering the probability of error, are the risk estimates not the same? Given this, the conclusions should also be reformulated (lines 267-268).
Response 2: Thanks for your very important comments. Following your suggestions, we have revised the sentence as following:
“The cerebrovascular disease mortality risk of PM2.5 to ethnic Han Chinese is slightly higher than ethnic Korean Chinese. However, the difference may also be affected by the confounding factors in the study, which needs to be confirmed by further research.”
Reviewer 2 Report
The manuscript has been revised according to my comments.
Author Response
Dear Reviewer,
Thank you very much for your work concerning our manuscript.